# The Basis of Diversity in Laminopathy Phenotypes Caused by Variants in the Intron 8 Donor Splice Site of the *LMNA* Gene

**DOI:** 10.3390/ijms26031015

**Published:** 2025-01-25

**Authors:** Olga Shchagina, Leisan Gilazova, Alexandra Filatova, Zulfiia Vafina, Aysylu Murtazina, Polina Chigvintceva, Olga Kudryashova, Aleksander Polyakov, Sergey Kutsev, Maria Bulakh, Mikhail Skoblov

**Affiliations:** 1Research Centre for Medical Genetics, 115522 Moscow, Russia; maacc@yandex.ru (A.F.); aysylumurtazina@gmail.com (A.M.); apol@dnalab.ru (A.P.); kutsev@mail.ru (S.K.);; 2Republic of Tatarstan Ministry of Healthcare Autonomous Public Healthcare, Institution Republic Clinical Hospital, 420064 Kazan, Russia; gafurova-lesya@mail.ru (L.G.); vzulfia@mail.ru (Z.V.); olga-10k@yandex.ru (O.K.)

**Keywords:** *LMNA*, laminopathia, cardiomyopathy, limb–girdle muscular dystrophy, lipodystrophy, splicing

## Abstract

Laminopathies are a broad spectrum of hereditary diseases caused by pathogenic variants of the *LMNA* gene. Such phenotypic diversity is explained by the function of intermediate filaments encoded by the *LMNA* gene. We examined a family with an overlapping phenotype of cardiac arrhythmia, cardiomyopathy, limb–girdle muscular dystrophy, and partial lipodystrophy. The cause of the disorder was a novel *LMNA*(NM_170707.4):c.1488+2T>C variant. The analysis of mRNA extracted from the probands’ blood showed a multitude of alternative splicing products, which was the cause of the complex phenotype in affected family members. Aside from that, we used minigene constructs to analyze the c.1488+2T>C variant, as well as other previously described variants affecting the same donor splice site in intron 8 (c.1488+1G>A, c.1488+5G>C, c.1488+5G>A). We demonstrated that these variants result in multiple splicing events, each producing splicing products with varying prevalence. Our experiments suggest that the variety of alternative transcripts contributes to complex phenotypes, while the quantitative ratio of these transcripts influences the varying severity of the disease.

## 1. Introduction

The *LMNA* gene (OMIM 150330) is located on chromosome 1q21.1–21.3 and encodes intermediate filaments of nuclear lamins A and C [1]. Pathogenic variants in the *LMNA* gene cause a broad spectrum of hereditary diseases called laminopathies [2]. The “muscle” phenotypes are the most common, such as Emery–Dreifuss muscular dystrophy types 2 and 3, congenital muscular dystrophy, dilated cardiomyopathy type 1A, and cardiac arrhythmias. Pathogenic variants in the *LMNA* gene are associated with peripheral nerve pathology (autosomal recessive Charcot–Marie–Tooth disease type 2B1) and adipose tissue pathology (familial partial lipodystrophy type 2, mandibuloacral dysplasia). In addition, pathogenic variants in the *LMNA* gene cause severe multisystemic disorders, such as Hutchinson–Gilford progeria and lethal restrictive dermopathy. Furthermore, it is known that different overlap phenotypes may occur.

Such phenotypic diversity is explained by the function of intermediate filaments encoded by the *LMNA* gene. Lamin A and the alternative splicing product lamin C are the key intermediate filaments of the inner nuclear membrane. Lamin A/C and lamin B form the framework of the inner membrane of the nucleus and hold the nuclear pores; lamin A/C is also able to directly bind chromatin, modulating its spatial organization. The interaction between chromatin and lamin A occurs in certain regions of the genome, defined as lamin-related domains (LADs) [3,4].

Three main pathophysiological hypotheses that do not exclude each other are proposed to explain the diversity of phenotypes associated with pathogenic variants in the *LMNA* gene [5]. (1) The structural hypothesis is explained by the role of lamins and related SUN proteins and nesprins of the LINC complex (linker of nucleoskeleton and cytoskeleton). These proteins support the mechanical integrity of the cell, linking the nucleoskeleton and cytoskeleton [6]. “Weakened” lamina can lead to a general loss of the ability of cells to withstand stress-induced damage, which can be crucial for tissues such as skeletal and cardiac muscles [7]. (2) The second hypothesis is based on the role of lamins as modulators of signal transduction pathways through specific interactions with chromatin and transcription factors such as Rb 27 (retinoblastoma) or SREBP-1 (sterol regulatory element-binding protein 1). It is assumed that pathogenic variants in the *LMNA* gene lead to modified or disrupted interactions within the lamin A/C signaling platform of the multi-protein complex, which contributes to the disruption of epigenetic modifications of chromatin and/or the disruption of various signaling pathways [8]. (3) The hypothesis of cellular toxicity suggests that the accumulation of mutated lamins has a dramatic toxic deleterious effect on the cells [9].

To date, approximately one hundred splice site variants in the *LMNA* gene have been described in the HGMD database [10]. The effect at the RNA and protein level has been studied for some splice site variants.

Most of the splice site variants located before exon 11 lead to “muscular” phenotypes, particularly cardiac arrhythmias/cardiomyopathies (72%) and muscular dystrophy (28%). In exon 11, six variants have been reported. Most of them lead to a progeroid phenotype, and one variant is associated with tight skin contracture syndrome [9,11,12,13], which is explained by the formation of truncated proteins and the accumulation of toxic products of the *LMNA* gene. This is a well-known pathogenic mechanism for Hutchinson–Gilford progeria, which was also described in cases with a recurrent synonymous c.1824C>T (p.G608=) variant. This pathogenic variant leads to the activation of a cryptic donor splice site and the deletion of 50 amino acids from prelamin [14]. Truncated prelamin A is called progerin and it cannot be processed into non-farnesylated mature lamin. The truncated and constantly farnesylated abnormal prelamin A is toxic for cells [15]. Other pathogenic variants, such as c.1821G>A (p.V607=), c.1822G>A (p.G608S), c.1868C>G (p.T623S), or c.1968G>A (p.Q656=), activate a similar mechanism [13,16,17,18].

The current study describes the clinical manifestation and molecular genetic effects of donor splice site variants on intron 8 of the *LMNA* gene.

## 2. Results

### 2.1. Case Reports

The family resides in Nizhnekamsk, Republic of Tatarstan, Russian Federation (Figure 1). Proband III.4, a 33-year-old female patient, was referred to a geneticist with complaints of leg weakness and gait disturbances. Her first symptoms, which appeared at the age of 25 years, included fatigue and shortness of breath during physical exercise. Her grandmother (I.2), father (II.2), and father’s siblings (II.1 and II.3) succumbed to sudden cardiac death in their fifth decade. Cardiac symptoms were also present in the proband’s cousins and their children. There were no consanguineous marriages in the family.

A neurological examination of the proband revealed hypotrophy and the weakness of proximal muscles in the upper limbs, thighs, and lower legs; these conditions were more prominent in the foot extensors. Other findings included bilateral foot drop, lumbar hyperlordosis, and waddling gait. Arm reflexes were reduced, and patellar and ankle reflexes were absent. Sensitivity was normal. There were no contractures or limitations in the range of movement in the spine and limb joints.

The creatine kinase level was elevated at 584 U/L. Needle electromyography showed myopathic changes in the limb muscles. Electrocardiography revealed a sinus rhythm with a heart rate of 58–83 bpm, pronounced bradyarrhythmia, a shortening of the PQ interval (0.10 s), and early ventricular repolarization syndrome. Echocardiography showed no abnormalities.

Molecular genetic analysis detected a single-nucleotide variant, c.1488+2T>C, in the *LMNA* gene (NM_170707.4). This variant affects the canonical donor splice site. The clinical data and the results of molecular genetic studies of other affected family members are summarized in Table 1.

As seen in Table 1, affected family members exhibited different clinical manifestations, which did not depend on the age at observation. The age of onset varied, with younger generations showing earlier manifestations, which was likely due to the more careful health monitoring of younger family members. The first signs of the disease in all examined affected family members were fatigue and shortness of breath when climbing stairs, which was indicative of cardiac arrhythmia. Subsequently, cardiomyopathy and weakness of limb–girdle muscles developed in some family members, without any joint stiffness or contractures. All affected family members, aged 12 to 43 years, had various heart problems, often involving rhythm disturbances. However, only some patients exhibited symptoms of skeletal muscle involvement.

It is interesting to note that patient III.11 had an abnormal adipose tissue distribution. The subcutaneous fat was normally distributed in the upper part of the body and was significantly reduced in the legs. The same pattern of adipose tissue distribution was observed in her older sister (III.5). The shoulder and pelvic girdle muscles were contoured, giving the impression of an athletic stature. In addition, patient III.5 had bilateral acanthosis nigricans (hyperpigmentation of the skin) in the area of ischial tuberosity. Their cousin III.1 had symptoms of moderate-to-severe diabetes mellitus type 2 (fasting blood glucose level ranging from 10.5 to 12.1 mmol/L). A similar metabolic disorder is usually observed in Dunnigan-type familial partial lipodystrophy type 2 (FPLD2).

### 2.2. Functional Analysis of LMNA c.1488+2T>C in a Patient with Dominant Cardiomyopathy

According to the ACMG guidelines, variation in canonical +/−1 or 2 splice sites likely leads to a disruption of splicing and should be classified as loss of function (LOF). However, sometimes there are no splicing abnormalities at these positions, and in some cases partial wild-type expression can be observed. Therefore, it is very important to carry out functional analysis of splicing variants in order not only to confirm the pathogenicity of the variant, but also to investigate the real mechanism of the variant’s influence on splicing.

We performed RT-PCR analysis of mRNA obtained from blood of patient III.11. The mRNA of her healthy brother III.10, who does not have this variant, was used as a control sample. Sequencing of the PCR product revealed the presence of the mutant isoform with a 9 bp deletion (Figure 2A). We were unable to obtain biomaterial from another family members for RNA research. Sequence analysis showed the presence of a weak cryptic donor site, which is activated when the WT donor site is broken. This in-frame deletion is located in the highly conservative lamin-tail domain (LTD) and results in a truncated protein (NP_733821.1:p.Val494_Thr496del). Lamins form the nuclear lamina, a protein net that lines the inner surface of the nuclear membrane. The main function of the nuclear lamina is to maintain the structural integrity of the nucleus. In addition, lamins play a role in binding chromatin-associated histones and other nuclear proteins. Structurally, lamins resemble other intermediate filaments (IFs), such as keratins and desmin, due to their α-helical rod central domain. However, unlike these other filaments, lamins are found exclusively in the nucleus and not the cytoplasm. Unlike other IFs, lamins form large filaments by intertwining their central domains, as well as interacting with other lamins and proteins through their carboxyl terminal segments—LTDs. Mutations in this region can cause phenotypes that go beyond muscular dystrophy [21].

Although cDNA samples were normalized before *LMNA* locus amplification, the amount of PCR product obtained for patient III.11 was higher than that for III.10. Unfortunately, we were unable to quantify the expression change due to the small number of original patient samples. However, targeted deep sequencing showed that the wild-type and abnormal transcripts were expressed at approximately the same level, indicating that the splicing change was not accompanied by a change in gene expression.

However, based on electrophoresis and Sanger data, we suggested that in addition to the two main transcripts, WT and Ex8 del9, others may exist. To investigate this, we carried out the high-coverage sequencing of RT-PCR products. As a result, we identified several less prevalent transcripts affecting exons 8 and 9 (Figure 2B): transcripts with the shortening of exon 8 and the retention of the acceptor part of the intron (Ex8 del9+IR20)—8.1%; transcripts with the shortening of exon 8 by 6 nucleotides (Ex8 del6)—6.7%; transcripts skipping exon 8—2.4%; and transcripts with the retention of the entire intron 8—1.9%. In total, these products comprise approximately the same number of transcripts that are formed from the wild-type allele, which indicates that they do not undergo degradation via the NMD mechanism.

### 2.3. Analysis of Splicing Variants in the LMNA Intron 8 Donor Site

Multiple variants have been described in the donor splice site of intron 8 (leading to different phenotypes) (Table 1).

To clarify the reason for the occurrence of different phenotypes, due to variants in the same splice site of *LMNA* intron 8, we carried out an in vitro minigene assay using constructs of c.1488+1G>A, c.1488+2T>C, c.1488+5G>C, and c.1488+5G>A.

For this approach, we created a minigene construct containing a locus with exons 8 and 9 of the *LMNA* gene and adjacent introns. Testing the minigene construct in HEK293T cells showed the correct splicing of exons 8 and 9, coinciding with the reference mRNA sequence. We introduced the studied variants by site-directed mutagenesis and carried out a minigene splicing assay. As a result, it was found that the variants led to the formation of several different transcripts—the main one, featuring a shortening of exon 8 by 9 nucleotides, and other transcripts—with intron retention and exon skipping.

To characterize these transcripts quantitatively and qualitatively, we carried out fragment analysis, as well as qPCR, and determined the relative prevalence of all isoforms originating due to different variants at the donor site. First, we demonstrated that each of the four variants leads to a decrease in the overall expression level of transcripts from minigene constructs. Such a phenomenon is known in the literature because transcription is associated with splicing, and changes in the course of splicing in different exons can lead to either an increase in or the complete absence of expression in general (Figure 3).

For the c.1488+2T>C variant, we found that 16.4% of the transcripts remained wild-type. This was generally confirmed by the RT-PCR data obtained from the patient, where the predominant expression of wild-type transcripts was observed from two alleles. SpliceAI also predicts a low probability of loss of the donor splice site (0.48). In the literature, there are descriptions of cases where canonical 5’ splice site (5’SS) GT>GC variants may be compatible with normal splicing [22].

Interestingly, for variants other than c.1488+1G>A, a residual number of wild-type transcripts are also observed. Thus, for the c.1488+5G>A variant, almost half of the transcripts remained intact, which corresponded well with the predictions of splicing changes. SpliceAI predicts this shortening of exon 8 by 9 nucleotides to be the most probable transcript, with a 0.62 score; however, the score for the loss of the site is 0.05. Indeed, the wild-type donor site of exon 8 has a high score (11.45) according to MaxEntScan, and the c.1488+5G>A variant weakens it, but not completely (5.43). The second variant in +5 position, c.1488+5G>C, leaves 12% of the wild-type transcript. SpliceAI and MaxEntScan prediction scores for this variant are lower, although not significantly. Only the c.1488+1G>A variant does not form wild-type transcripts, and indeed, it has a very high SpliceAI score for site loss (0.99).

The second most prevalent splicing alteration is the shortening of exon 8 by 9 nucleotides, which also corresponds with the RT-PCR data for the patient’s sample. This transcript is formed in all splicing variant cases in the current study; its prevalence varies from 28.5% to 62.5%. Taking into account the dominant nature of the inheritance of this disease, it can be assumed that the shortening of exon 8 by 9 nucleotides plays a key role in molecular pathogenesis.

## 3. Discussion

Since the discovery of the *LMNA* gene, variants in which cause laminopathies, over 1000 scientific and clinical studies have been published. These studies were aimed at establishing causality between morphologic/functional defects of cells with laminopathy and the heterogeneous clinical phenotypes of this disease group. It has been shown that variants affecting the α-helical rod domain led to the disruption of structural function of lamin A and C intermediate filaments, causing phenotypes of muscular dystrophy, cardiomyopathy, and neural amyotrophy. Hegele R. showed that the development of these laminopathy types is 8.4 times more likely to be caused by pathogenic variants located before the NLS point (*p* < 0.0001) [23]. The variants affecting the carboxyl tail (β-immunoglobulin-like-fold), which enables the interaction between lamin A and non-nucleoskeletal elements (such as DNA, chromatin, and/or transcription factors), have an impact on gene expression and lead to phenotypes characterized by metabolic dysfunction and ageing: lipodystrophy, progeria, and lethal restrictive dermatopathy.

We examined a family with a disease caused by a heterozygous c.1488+2T>C variant in the *LMNA* gene. This variant was not present in the population databases such as gnomAD (https://gnomad.broadinstitute.org/, accessed on 7 January 2020) (PM2 ACMG [24], in silico prediction programs evaluate it to be pathogenic (meta scores = 8 (very strong) and individual predictions = 17 (Very Strong)) accor ding to VarSome [25] (PP3 ACMG). The c.1488+2T>C variant leads to the disruption of exon 8 splicing; however, the majority of the product from a damaged allele probably escapes NMD due to the fact that deletions occur within the reading frame. Therefore, we cannot apply the PVS1 criterion and must use the PM4 (modified protein length synthesis) criterion ACMG. Additionally, to assess pathogenicity in this case, we use the family segregation criterion as a moderate (PM) one due to the abundance of family information available. According to the ACMG criteria, after all the studies conducted, the previously undescribed variant c.1488+2T>C can be classified as probably pathogenic (PM2, PM4, PP3, and PP1-M).

The total number of affected family members was 12, of whom 9 were examined. Comparing these affected family members, we noticed significant differences in their clinical presentations and severity. These differences did not solely depend on the age at examination. Three adult patients, aged 27, 34, and 43, exhibited only cardiac disturbances, while three others, aged 19, 34, and 35, also had weakness or atrophy of skeletal muscles. One patient additionally displayed symptoms of Dunnigan-type familial partial lipodystrophy. We presume that the overlapping phenotypes could be caused by the presence of different RNA products, predominantly the RNA isoform with a 9-nucleotide shortening of exon 8, which leads to the truncation of the protein NP_733821.1:p.Val494_Thr496del. Previously, the authors of a cohort study [26] described a p.Thr496Met (c.1487C>T) variant in a patient with arrhythmia of an uncertain origin. In addition, two deletions were described. These were c.1483del, leading to a frameshift and therefore p.Val495* stop codon formation in a patient with arrhythmia [27], and c.1488_1488+9del, which was found in a patient with congenital muscular dystrophy [19,28]. No other variants affecting amino acid residues 494–496 were reported. The aforementioned isoform comprised 26.4% of all transcripts. Aside from that, length alterations without a frameshift can be caused by other registered isoforms: one with the shortening of exon 8 by 6 nucleotides (Ex8 del6) (7.1%) and another with the retention of the entire intron 8 (1.8%). These three isoforms comprised 35% of all transcripts. In HGMD [10], there is information on 31 minor deletions and insertions in the reading frame, all leading to “muscle” phenotypes: 10 cases of cardiomyopathy and 21 cases of various muscular dystrophy types (Emery–Dreifuss and limb–girdle).

It is worth noting that arrhythmia is frequently caused by pathogenic variants, leading to haploinsufficiency; cardiomyopathy develops later; and the skeletal muscular dystrophy phenotype may not manifest. Wolf CM et al. conducted an experiment on mice with a knocked-out *LMNA* gene. Despite having one normal allele, mice with Lmna(+/−) developed cardiac abnormalities. Conductive system function was normal in neonatal Lmna(+/−) mice; however, at the age of 4 weeks, the atrioventricular (AV) node myocytes had abnormal nuclear shapes and signs of apoptosis. The in vivo telemetric and electrophysiological examination of 10-week-old Lmna(+/−) mice showed the defects of AV conductivity and both atrial and ventricular arrythmia. These were similar to the corresponding findings in human patients with heterozygous *LMNA* pathogenic variants. Isolated myocytes of 12-week-old Lmna(+/−) mice demonstrated contractility impairment. Heart examination in vivo in aged mice with Lmna(+/−) revealed dilated cardiomyopathy, in some cases without an apparent conductive system disorder. However, neither the results of histological examination nor creatine phosphokinase activity level in blood serum showed any skeletal muscle pathology. Thus, the experiments showed that lamin insufficiency affects node myocytes the most, while non-conductive myocytes in the heart or in skeletal muscles are less susceptible [29]. Pathogenic variants in other cytoplasmic intermediate filaments like desmin (DES) cause cardiomyopathies, for example, non-compaction cardiomyopathy. It is also known, that splice site mutations in *DES* cause cardiomyopathies [30,31]. However, the pattern of affected muscles and the symptoms of lipodystrophy in some of the patients we examined suggested the LMNA gene as the cause of the disease.

The most probable cause of lipodystrophy symptoms is the detected RNA variant with the skipping of exon 8 (skipping ex8), comprising 2.5% of all transcripts. The deletion of the entire exon 8 does not lead to a frameshift: 36 amino acid residues are lost, among them the interaction «hot spot» of the lamin sterol regulatory-binding protein, and the effect of this deletion is similar to the effect of common lipodystrophy variants in codon 482 of the *LMNA* gene. This codon is the «hot spot» for the Dunnigan-type familial partial lipodystrophy phenotype, and its missense variants—p.R482Q, p.R482W, and p.R482L—have been described as the cause of FPLD2 [23]. This pathogenic variant may cause damage to adipocytes due to abnormal interactions with transcription factors, and disrupt the spatial organization of the affected LMNA protein (not in the nuclear periphery, but chaotically in the cytoplasm).

To establish the nature of the differences in phenotypes caused by variants in the same donor splice site in intron 8 *LMNA*, we carried out in vitro minigene analysis with constructs including the following variants: c.1488+1G>A, c.1488+2T>C, c.1488+5G>C, and c.1488+5G>A.

The results of minigene construct analysis for the c.1488+2T>C variant corresponded with the analysis results for the proband’s RNA extracted from peripheral blood.

The c.1488+1G>A variant was described in a study including patients with a muscular dystrophy phenotype in a sporadic case of Emery–Dreifuss muscular dystrophy in a young male patient. At the age of 24 years, the patient underwent surgery to implant a pacemaker because of pronounced arrhythmia [32]. The major form of the mRNA construct with the c.1488+1G>A variant was a product with the 9 bp deletion; the products with the intron 8 retention and the exon 8 deletion were less prevalent. Seeing as the total RNA quantity was decreased by more than 20% compared to the wild-type, we can assume that it is likely caused by the presence of isoforms undergoing NMD. It is also worth noting that c.1488+1G>A is the only variant that leads to the complete absence of the normal splicing product, which corresponds with the early onset age of laminopathy.

The c.1488+5G>C variant in intron 8 has been described in patients with Dunnigan-type partial lipodystrophy [20,33]. This pathogenic variant leads to the formation of truncated lamin isoforms. The functional analysis of this variant has shown that truncated lamin partially undergoes NMD. This particular effect of the variants likely explains the arrhythmia seen in affected patients. At the same time, the described symptoms of mild limb–girdle muscular dystrophy, noted in two out of three probands with the c.1488+5G>C variant, can be explained by the presence of transcripts shortened by 9 bp and elongated by 84 bp because of intron 8 retention. As previously described, variants affecting the α-helical rod domain led to the disruption of structural function, causing phenotypes of muscular dystrophy, cardiomyopathy, and neural amyotrophy [2]. Deletions and insertions that do not shift the reading frame do not cause product degradation, but they can significantly affect the protein structure. However, the predominant phenotype in the patients was partial lipodystrophy. Researchers have carried out the immunofluorescent microscopy of cells with normal and mutant lamin and showed the partial presence of the protein in case of the c.1488+5G>C variant. However, similarly to the effect of the p.R482W pathogenic variant [34], which is common in cases of lipodystrophy, this protein is not located in the periphery of the nucleus, but distributed chaotically, which can lead to decreases in expression and explain the partial Dunnigan-type lipodystrophy phenotype. These results correspond with our data on the presence of transcripts with a deletion of the entire exon 8, which leads to a loss of the lamin sterol regulatory-binding protein interaction site.

The c.1488+5G>A variant was present in ClinVar [https://www.ncbi.nlm.nih.gov/clinvar/RCV000057306.1/, accessed on 7 January 2020]; however, no information on the clinical observations or analyzed tissue was given. The increased level of the p.494_496del product allows us to assume that the variant would lead to a muscular dystrophy phenotype with a less severe phenotype.

In summary, a comparison of the clinical presentation of our patients with previously reported cases of similar variants suggests that the presence of different components of the muscle phenotype (arrhythmias, cardiomyopathy, muscular dystrophy), as well as the severity of laminopathy, may depend on the amount of the wild-type isoform. This is typical of variants affecting splicing sites. Thus, the patient with the c.1488+1G>A variant, which leads to the complete loss of the wild-type isoform, exhibited more severe and earlier cardiac and skeletal muscle features, similar to cases with nonsense variants [10]. At the same time, patients with the variant c.1488+2T>C and those with the variant c.1488+5G>C, which did not differ significantly in effect, exhibited diverse clinical presentations, including metabolic disorders and features of the lipodystrophy phenotype, but with later cardiac involvement.

The transcript with the deletion of exon 8, which leads to the loss of the active center of interaction between lamin and the factors of lipid metabolism regulation, is present in all examined nucleotide sequence variants. This likely signifies that the quantitative ratio of this transcript’s product and other protein forms in cells plays a key role in the realization of the Dunnigan-type lipodystrophy phenotype.

Different phenotypes and clinical severity in family members with the same pathogenic variant can possibly be explained by both the duration of the disease and also by the effect of genetic surroundings on the course of splicing and the prevalence of various mRNA products. However, this hypothesis requires further experimental confirmation.

## 4. Materials and Methods

The materials for this study were DNA and RNA samples extracted from the whole blood of affected and healthy members of one family (the pedigree is presented in Figure 1). This study was approved by the local ethics committee of the Federal State Budgetary Institution “Research Centre for Medical Genetics” (the approval number 2018-5/3). Informed consent was obtained from all subjects involved in the study.

The family resides in Nizhnekamsk, Republic of Tatarstan, Russian Federation. DNA was extracted from whole blood samples using a Wizard^®^ Genomic DNA Purification Kit (Promega, Madison, WI, USA) according to the manufacturer’s protocol.

Automated Sanger sequencing was carried out using an ABI Prism 3100×L Genetic Analyzer (Applied Biosystems, Foster City, CA, USA) according to the manufacturer’s protocol. Primer sequences were chosen according to the NM_170707.4 reference sequence.

(EX1F1-GGGCAGCGCTGCCAACCTGC; EX1R1-CCTCTTCAGACTCGGTGATGC; EX1F2-CGGTCTACATCGACCGTGTGC; EX1R2-CTCCACTCCCCGCCAGGCAC; EX2F-ACAGACTCCTTCTCTTAAATCTAC; EX2R-GTGTACATGTGTTAGGTGGGGC; EX3F-TTCTGTGACCCCTTTTCCTCATC; EX3R-ATCACCCAGCCCCAGCCCTC; EX4F-CTAATTCTGATTTTGGTTTCTGTGTC; EX4R-GTGGGTAAGGGTAGGGCTGCC; EX5F-TGCCTCCACCCCTCCCAGTCAC; EX5R-ATCCGGCCCAGACTCTAGGCC; EX6F-CCCTTGGGAGCTCACCAAACC; EX6R-ACTGCCAGCACCTCGGCGAC; EX7F-CTGGCCTTGACTAGACCCCCAC; EX7R-GCAGCTGTATCCCCTTAGACCC; EX8F-CACCCAAGAGCCTGGGTGAGC; EX8R-CCATCGACACCCAAGGTCTCC; EX9F-GTGTCAGGGCGCTTGGGACTC; EX9R-GGGAGCCTCGTCCAGCAAGC; EX10F-CCTGGCCCTGACCCTTGGAC; EX10R-CCAGGCCAGCGAGTAAAGTTCC; EX11F1-AGTGGTCAGTCCCAGACTCGC; EX11R1-AAGCTGCCACCCCCACTGCC; EX11F2-TCTGCCTCCAGTGTCACGGTC; EX11R2-CGCCTGCAGGATTTGGAAGAC; EX12F-CCAGGCCCCTGTTGTTCACAC; EX12R-GTGTTTTTCCTTCAGTATAAAACCAC).

RT-PCR analysis of the patients’ RNA samples

To obtain mRNA, we separated PBMCs from whole blood using a density gradient centrifugation method with Ficoll. Then cells were lysed in ExtractRNA buffer (Evrogen, Moscow, Russia) and total RNA was prepared by phenol–chloroform extraction. For purification from genomic DNA, RNA was treated with DNase I (Thermo Fisher, Waltham, MA, USA). cDNA was obtained using a reverse transcription system (DIALAT Ltd., Moscow, Russia). To assess the quality of the cDNA, qPCR was carried out using primers for the B2M housekeeping gene.

PCR amplification was carried out using the following primers: LMNA67JUNC_F (5′ GGGCGAGGAGGAGAGGCTAC) and LMNA970JUNC_R (5′ GCGCATGGCCACTTCTTCCC). The PCR products were analyzed on 2% agarose gel with subsequent Sanger sequencing.

The PCR products were sequenced using NGS. NGS libraries were prepared and sequenced on an Ion Torrent S5 (with coverage of approximately 5000). The raw sequencing data was processed with a custom pipeline based on the open-source bioinformatics tools STAR 2.7.8a, Samtools 1.3.1, and SAJR (no version number). Splice junctions were visualized using the Sashimi plot in IGV.

In vitro splicing analysis with a minigene assay

In vitro splicing analysis was carried out using a minigene expression system, as described before [35]. For minigene construction, the region encompassing exons 8 and 9 of the *LMNA* gene was amplified from genomic DNA using the following primers: LMNA-F-XhoI (5′ AAAACTCGAGATGGAAGGAGAGGCCTCAAT) and LMNA-R-BamHI (5′ AAAAGGATCCGAACTCATTTCCACCCCAGA). The obtained wild-type PCR product was cloned into the pSPL3-Flu2 plasmid vector (lab made). The structure of minigenes was confirmed by Sanger sequencing. To introduce the studied variant, we used the Single-Primer Site-Directed Mutagenesis Method [36].

Human embryonic kidney 293T (HEK293T) was cultured in high glucose Dulbecco’s modified Eagle’s medium (DMEM) with Alanyl Glutamine (PanEco company, Berg am Irchel, Switzerland) supplemented with 10% FBS (HyClone), 100 U/mL penicillin, and 100 μg/mL streptomycin. Cells were seeded at 7 × 10^4^ cells/well in 24-well poly-L-lysine-coated plates 24 h before transfection. We performed the transfection of minigene plasmids (500 ng/well) with control, using the Calcium Phosphate method for the HEK293T cell line [37].

Some 48 h after transfection, total RNA was isolated from the transfected cells using the ExtractRNA reagent (Evrogen), following the manufacturer’s recommendations. RNA was treated with DNase I (Thermo Fisher Scientific) and reverse-transcribed using the ImProm-II™ Reverse Transcription System (Promega).

To detect the splicing products, we used a plasmid-specific primer TurboFP-F (ACAAAGAGACCTACGTCGAGCA) and an *ADSL*-gene-specific primer Ex9R (TGGGATTCCGCTTATATGGCA). The PCR products were analyzed in polyacrylamide gel with urea, and then subjected to Sanger sequencing. To assess the ratio of mRNA isoforms, fragment analysis with the FAM-6-tagged forward primer was conducted.

qPCR reactions were carried out with an EvaGreen qPCR Master mix (Biotium, Fremont, CA, USA) and 250 nM of each primer (LMNA-7F:GTCACCAAAAAGCGCAAACT and LMNA-10R:GGTGATGGAGCAGGTCATCT). The efficiency and specificity of primers were assessed via a standard curve obtained by the serial dilutions of cDNA. The relative expression level of LMNA minigene constructions compared to Neomycin gene expression was calculated by the ΔΔCt method. All experiments were conducted with a minimum of three biological replicates, and each contained a minimum of three technical replicates. Statistical analysis was carried out via two-sided Student’s *t* test using R-4.4.2 for Windows software. Data were presented relative to the WT construct as mean ± SEM among biological replicates.

## 5. Conclusions

We showed that splice site variants can realize the effect via various mechanisms; therefore, during the detection of a variant that affects the splice site, we may expect different phenotypic effects caused by the loss or functional alteration of the protein products. All of the variants examined in the current study caused a disruption in the *LMNA* intron 8 donor splice site. The «hot spot» of Dunnigan-type lipodystrophy is located in exon 8; therefore, the examined patients with the c.1488+2T>C variant and the patients described in the literature with the c.1488+1G>A and c.1488+5G>C variants have a wide spectrum of laminopathy symptoms, such as skeletal and cardiac muscle lesions, arrhythmia, and partial lipodystrophy symptoms.

## Figures and Tables

**Figure 1 ijms-26-01015-f001:**
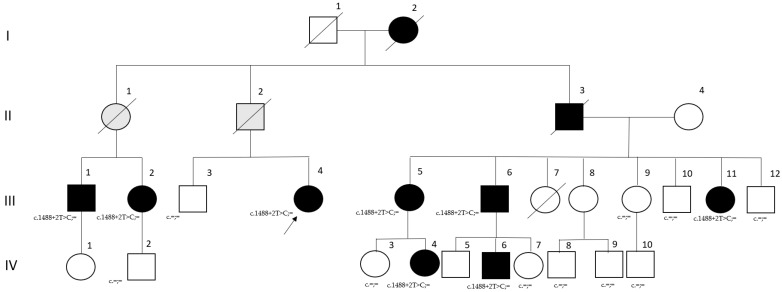
The pedigree of the examined family. The arrow indicates the proband; black shading symbols indicate affected family members who were examined; grey symbols—unexamined family members with cardiac history; white symbols—healthy family members, the crossed-out symbol represents a deceased family member. The results of genotyping of family members are indicated (c.1488+2T>C;=—carriers; c.=;=—not carriers).

**Figure 2 ijms-26-01015-f002:**
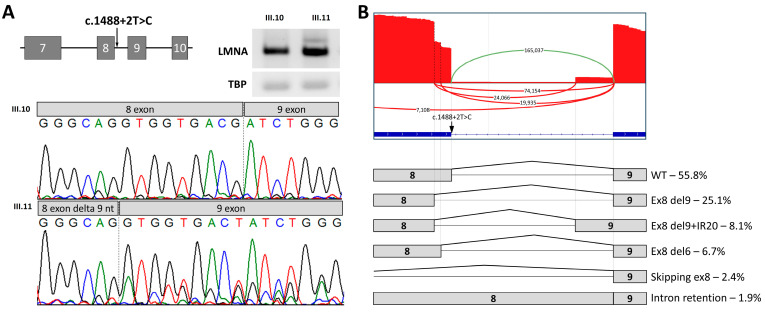
The c.1488+2T>C variant leads to the disruption of exon 8 splicing. (**A**) The RT-PCR analysis of LMNA mRNA from PBMCs of patient III:11 (carrier of the variant) and her healthy brother III.6. The Sanger sequencing results of reference and aberrant (c.1488+2T>C) PCR products demonstrated a heterozygous mutant isoform with a 9 bp shortening of exon 8. The weak larger PCR product observed in III.11 represents an intron retention isoform. We were unable to Sanger sequence it separately, but it was of suitable size and we confidently identified it by deep sequencing, allowing us to quantitatively identify all minor isoforms. (**B**) A Sashimi plot used to visualize splice junctions for the c.1488+2T>C RT–PCR product with the quantification of the resulting isoforms.

**Figure 3 ijms-26-01015-f003:**
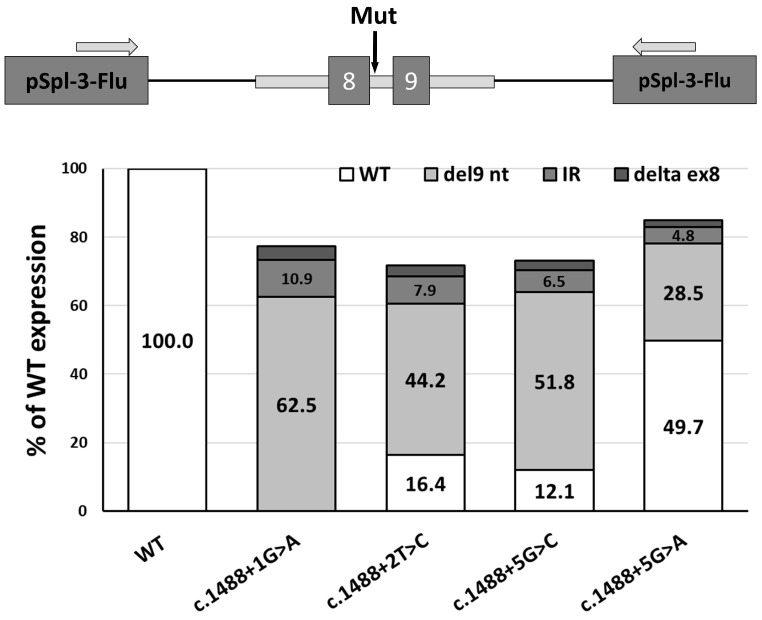
Minigene analysis of splicing variants in the donor site of *LMNA* exon 8. The histogram shows that the quantitative assessment data for isoforms originated as a result of the damage different splicing variants caused to the donor site.

**Table 1 ijms-26-01015-t001:** Clinical observations in affected family members (f—female, m—male, n/a—not available, Abs.—absent).

Number	Gender	Age of Onset	Age at the Time of Examination	Cardiac Involvement	Skeletal Muscle Involvement	Other Findings	**CK (U/L)**	***LMNA* Genotype**
II.1	f	n/a	Died at the age of 58 years	Cardiac arrhythmias. Implanted pacemaker. Acute heart failure.	n/a	n/a	n/a	n/a
II.2	m	n/a	Died at the age of 57 years	Cardiac arrhythmias. Implanted pacemaker. Acute heart failure.	n/a	n/a	n/a	n/a
II.3	m	n/a	Died at the age of 47 years	Acute heart failure.	Muscle weakness	n/a	n/a	n/a
III.1	m	n/a	43	Bradysystolic atrial fibrillation (40–43 bpm). Polymorphic ventricular extrasystole. Left ventricle hypertrophy. Implanted pacemaker.	Normal	Diabetes mellitus type 2, moderate severity	263	c.1488+2T>C;=
III.2	f	n/a	34	Arrhythmia.	Normal	Abs.	824	c.1488+2T>C;=
III.4	f	25	35	Bradyarrhythmia. Shortening of PQ interval (0.10 s). Early repolarization syndrome. Nonspecific intraventricular conduction disturbance.	Gait disturbances, atrophy of proximal muscles of upper limbs, thighs and calves	Abs.	584	c.1488+2T>C;=
III.5	f	27	34	Persistent atrial fibrillation. Ventricular extrasystole. Hypertrophic cardiomyopathy.	Leg muscle weakness (foot drop, Gowers’ sign)	Lack of subcutaneous fat in the lower extremities and the shoulder girdle. Bilateral acanthosis nigricans.	210	c.1488+2T>C;=
III.6	m	19	27	Paroxysmal atrial fibrillation. Tachyarrhythmia (98 BPM). Impaired myocardial contractility of LV.	Normal	Abs.	372	c.[1488+2T>C];[=]
III.11	f	16	19	Supraventricular/ ventricular extrasystole. Tachycardia. Cardiomegaly.	Leg muscle weakness and atrophy	Abnormal subcutaneous fat distribution: normal in the upper part of the body, significantly reduced in the legs.	317	c.1488+2T>C;=
IV.4	f	10	12	Bradyarrhythmia	Normal	Normal	330	c.1488+2T>C;=
IV.6	m	not manifested	6	Normal	Normal	Congenital malformations, cheiloschisis	136	c.1488+2T>C;=
Patients from the articles [19,20]	m	n/a, diagnosis at the age of 17	Died in his early 30 s (death possibly unrelated to EDMD)	Pacemaker at age 24 which was replaced at age 30	Patient needed a cane and a scoot for ambulation	n/a	n/a	c.1488+1G>A;=
Patients from the articles [19,20]	f	5	20	The prolongation of the QT interval with diffuse repolarization abnormalities (14 years)	Normal	Acanthosis nigricans, hypertension, hypertriglyceridemi, hyperinsulinemia, diabetes mellitus type 2, facial and body hirsutism, male body habitus, severe lipodystrophy.	n/a	c.1488+5G>C;=
Patients from the articles [19,20]	f	17	26	Left ventricular strain pattern	Proximal muscle weakness of both upper and lower extremities	Hypertriglyceridemia, hypertension, diabetes mellitus type 2, light-brown papillomatous hypertrophic plaques, verrucous hyperkeratosis, numerous skin tags over neck and axillae, severe lipodystrophy.	Normal	c.1488+5G>C;=

## Data Availability

The original contributions presented in this study are included in the article. Further inquiries can be directed to the corresponding author.

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
