# Peer review of "The Basis of Diversity in Laminopathy Phenotypes Caused by Variants in the Intron 8 Donor Splice Site of the LMNA Gene"

_ijms, 2025, doi:10.3390/ijms26031015_

Round 1
Reviewer 1 Report
Comments and Suggestions for Authors
The study by Shchagina and co-authors identifies a splice variant in LMNA in a family with overlapping phenotypic characteristics of cardiomyopathy, arrhythmias, limb-girdle muscular dystrophy and partial lipodystrophy. The authors determine that this variant results in the production of multiple isoforms that affect LMNA exon 8. In addition, this variant along with other splice variants affecting the same intron 8 donor splice site are analyzed in a cell culture system, determining differences in expression levels of resultant transcripts. This is an interesting study, attempting to clarify the complex phenotypes caused by these LMNA splice variants and their influence on varying disease severity.
Comments on this manuscript:
1) The majority of the proband’s family members have the same LMNA variant (c.1488+2T>C, as shown in Table 1), however they exhibit a wide range of phenotypes. It is still unclear how this may occur, how do the authors explain this? Could it be associated with differences in expression levels (%) of the different isoforms or reduction in WT levels among the various members? Apart from III.7 and III.6 (normal control), have the authors checked by RT-PCR and sequencing any of the other family members? This could be useful to address the above question.
2) In the RT-PCR image of figure 2, there appears to be a faint but larger sized product in III.7. Have the authors purified and sequenced this product and do they know which isoform it corresponds to? Is this the intron retention isoform? Also, a housekeeping gene should be included as a control in this figure.
3) While RT-PCR is only semi-quantitative, the image in Figure 2A implies that there is increased expression of exon 8-9 in III.7 compared to III.6. However, based on findings from Figure 3, this is actually not true and in fact the variant shows ~20% decrease in expression compared to WT. How do the authors explain this apparent contradiction?
4) Along those lines, it is unclear why there is an overall 20% decrease in lamin A expression observed by all variants analyzed in the cell culture system. Please explain.
5) Given that the patient is heterozygous for the c.1488+2T>C variant, I assume that 50% of LMNA product would be WT and 50% (or less, based on Figure 3 data) would be Ex8 splice isoforms. If this is correct, then why does the percentage of all splice variants predicted by Sashimi plot add up to a total of 101.9%?
6) In the discussion, page 4 lines 259-261, the authors claim that arrhythmia in the patients is due to the presence of the splicing product Ex8 del9+IR20 (8.6%) but it is unclear how they reached this conclusion. Please explain.
7) Similarly, on page 5 lines 292-294, it is unclear how the authors reach the conclusion that transcripts shortened by 9bp and elongated by 84bp explain the mild limb-girdle muscular dystrophy phenotype. Please clarify.
8) In contrast to the above (points 6 and 7), on page 5 lines 309-318, the authors propose that the phenotype and clinical severity of laminopathy is mostly determined by the WT isoform rather that the 9bp shorter transcript. This contradicts the previous hypotheses and makes it unclear as to whether it is downregulation of the WT transcript or the presence of the variant isoforms that causes disease.
9) The authors mention that the identified variant c.1488+2T>C is located in the lamin-tail domain of the protein. It would be useful to include in the discussion section some information regarding the functional role of this domain and how its disruption could lead to disease. Along those lines, a diagram showing lamin A domains, their functional significance and the location of the different variants mentioned within the text (published and analyzed in this study) would be helpful.
10) A schematic diagram depicting the different intron 8 variants and their resultant splice isoforms would be very helpful to the reader. Inclusion of the proposed phenotypes associated with the various isoforms would be a useful addition.
11) Page 2, line 184: ‘(PMID)’, this reference is not cited properly
Author Response
We thank Reviewer 1 for an exceptionally attentive review of our manuscript. We enjoyed preparing the answers. Thank you for the identified errors and inaccuracies that we made during the formation of the first draft of the manuscript.
Comments 1: The majority of the proband’s family members have the same LMNA variant (c.1488+2T>C, as shown in Table 1), however they exhibit a wide range of phenotypes. It is still unclear how this may occur, how do the authors explain this? Could it be associated with differences in expression levels (%) of the different isoforms or reduction in WT levels among the various members? Apart from III.7 and III.6 (normal control), have the authors checked by RT-PCR and sequencing any of the other family members? This could be useful to address the above question.
Response 1: Indeed, it would be very interesting to investigate splicing disruption in other carriers of the c.1488+2T>C variant in this family. However, the family is not very sociable, and we were unable to obtain biomaterial from them for research. It should be noted that one of the common mechanisms that can influence splicing differently is the presence of complex alleles, which we did not find when conducting DNA diagnostics of family members. We therefore suggested that the observed wide range of phenotypes among family members is due to their different genetic backgrounds, as we noted in the discussion section. We have explained this fact in the text (highlighted in green).
Comments 2 In the RT-PCR image of figure 2, there appears to be a faint but larger sized product in III.7. Have the authors purified and sequenced this product and do they know which isoform it corresponds to? Is this the intron retention isoform? Also, a housekeeping gene should be included as a control in this figure.
Response 2 Indeed, the weak larger PCR product observed in III.7 represents an intron retention isoform. We were unable to Sanger sequence it separately, but it is of suitable size and we confidently identified it by deep sequencing, allowing us to quantitatively identify all minor isoforms. We also added to Figure 2 the electrophoresis results after amplification of the TPB gene locus as a housekeeping gene (highlighted in green).
Comments 3 While RT-PCR is only semi-quantitative, the image in Figure 2A implies that there is increased expression of exon 8-9 in III.7 compared to III.6. However, based on findings from Figure 3, this is actually not true and in fact the variant shows ~20% decrease in expression compared to WT. How do the authors explain this apparent contradiction?
Response 3 Although cDNA samples were normalized before LMNA locus amplification, the amount of PCR product obtained for patient III.7 was higher than that for III.6. Unfortunately, we were unable to quantify the expression change due to the small number of original patient samples. However, targeted deep sequencing showed that the wild type and abnormal transcripts were expressed at approximately the same level, indicating that the splicing change was not accompanied by a change in gene expression. We have explained this fact in the text (highlighted in green).
Comments 4 Along those lines, it is unclear why there is an overall 20% decrease in lamin A expression observed by all variants analyzed in the cell culture system. Please explain.
Response 4 Indeed, in vitro experiments using the minigene system showed that the studied variants lead to an overall decrease in the expression of chimeric transcripts. The link between transcription and splicing has been shown especially for the first exons of genes. In the case of the our minigenes, exons 8 and 9 of the LMNA gene turned out to be the second and third in order, which could have influenced the decrease in expression. However, it requires more detailed study. We believe that the obtained data on the ratio of abnormal transcripts are relevant, as they are in good agreement with the data obtained from targeted deep sequencing of the patient sample.
Comments 5 Given that the patient is heterozygous for the c.1488+2T>C variant, I assume that 50% of LMNA product would be WT and 50% (or less, based on Figure 3 data) would be Ex8 splice isoforms. If this is correct, then why does the percentage of all splice variants predicted by Sashimi plot add up to a total of 101.9%?
Response 5 Thanks to the reviewer for carefully studying our article. Indeed, we made a mistake in calculating the percentage of all splicing options. We have corrected the data both in the article and in the figure. (highlighted in green)
Comments 6 In the discussion, page 4 lines 259-261, the authors claim that arrhythmia in the patients is due to the presence of the splicing product Ex8 del9+IR20 (8.6%) but it is unclear how they reached this conclusion. Please explain.
Response 6 We thank the Reviewer for his attention. This phrase is from an old version of the article. We deleted it.
Comments 7 Similarly, on page 5 lines 292-294, it is unclear how the authors reach the conclusion that transcripts shortened by 9bp and elongated by 84bp explain the mild limb-girdle muscular dystrophy phenotype. Please clarify.
Response 7 We have tried to clarify our opinion in the text of the manuscript (highlighted in green).
Comments 8 In contrast to the above (points 6 and 7), on page 5 lines 309-318, the authors propose that the phenotype and clinical severity of laminopathy is mostly determined by the WT isoform rather that the 9bp shorter transcript. This contradicts the previous hypotheses and makes it unclear as to whether it is downregulation of the WT transcript or the presence of the variant isoforms that causes disease.
Response 8 Thanks to the reviewer. We have rewritten this paragraph so that our opinion is clearer. (highlighted in green)
Comments 9 The authors mention that the identified variant c.1488+2T>C is located in the lamin-tail domain of the protein. It would be useful to include in the discussion section some information regarding the functional role of this domain and how its disruption could lead to disease. Along those lines, a diagram showing lamin A domains, their functional significance and the location of the different variants mentioned within the text (published and analyzed in this study) would be helpful.
Response 9
Thanks to the reviewer for your interesting question. In the text of our manuscript, we have tried to highlight the uniqueness of the lMNA/C tail domain. Many papers and studies have been published on the structure of these proteins' domains, so we do not want to overload the manuscript with additional information. Protein scientists are better equipped to discuss this topic in detail, and we believe it is important to focus on other aspects of our research.
It is worth noting that the location of the LTD and its interaction with other proteins play a crucial role in explaining the unique lipodystrophy and progeria phenotypes associated with LMNA mutations. In contrast, the IF desmin are more closely linked to muscle-related phenotypes.
Comments 10 A schematic diagram depicting the different intron 8 variants and their resultant splice isoforms would be very helpful to the reader. Inclusion of the proposed phenotypes associated with the various isoforms would be a useful addition.
Response 10 The schematic diagram is shown in the updated Figure 2.
Comments 11 Page 2, line 184: ‘(PMID)’, this reference is not cited properly
Response 11 It has been fixed.
Reviewer 2 Report
Comments and Suggestions for Authors
In the original manuscript "The basis of diversity in laminopathy phenotypes caused by variants in the intron 8 donor splice site of the LMNA gene" submitted by Shchagina et al. to IJMS, the authors identified a LMNA splice site mutation in a Russian family. The topic of this manuscript is interesting to a broad readership. However, I suggest several changes /corrections, before this manuscript should be published:
1.) I would write LMNA in the title in Italics, since you use the gene name.
2.) Figure 1: I would indicate with +/- symbols which family members carry the mutation.
3.) Figure 2: I would also change the numbering of the individuals starting from right to left (1-8). In the presented figure this is mixed and confusing. Be careful, that you change than also the numbering in the text and in the table according to these corrections.
4.) Table 1 needs also some correction. You insert here reference [19] and [20] instead of indicating the individual . This should be changed!
5.) In the discussion, I would also mention that mutations in other cytoplasmic intermediate filaments like desmin cause cardiomyopathies. For example, it is known, that DES mutations cause non-compaction cardiomyopathy (see 'Noncompaction cardiomyopathy is caused by a novel in-frame desmin (DES) deletion mutation within the 1A coiled-coil rod segment leading to a severe filament assembly defect’ (2019). It is also known, that splice site mutations in DES cause cardiomyopathies (‘The Desmin Mutation DES-c.735G>C Causes Severe Restrictive Cardiomyopathy by Inducing In-Frame Skipping of Exon-3’ (2021)). Both points I would mention and discuss in the context of a comparison of LMNA and DES.
However, I think that the authors can fix these points. Therefore, I suggest a minor revision. Good luck!
Author Response
In the original manuscript "The basis of diversity in laminopathy phenotypes caused by variants in the intron 8 donor splice site of the LMNA gene" submitted by Shchagina et al. to IJMS, the authors identified a LMNA splice site mutation in a Russian family. The topic of this manuscript is interesting to a broad readership. However, I suggest several changes /corrections, before this manuscript should be published:
- I would write LMNA in the title in Italics, since you use the gene name.
Fixed
- Figure 1: I would indicate with +/- symbols which family members carry the mutation.
Information about the carrier has been added to the pedigree
- Figure 2: I would also change the numbering of the individuals starting from right to left (1-8). In the presented figure this is mixed and confusing. Be careful, that you change than also the numbering in the text and in the table according to these corrections.
Fixed
- Table 1 needs also some correction. You insert here reference [19] and [20] instead of indicating the individual . This should be changed!
Fixed
- In the discussion, I would also mention that mutations in other cytoplasmic intermediate filaments like desmin cause cardiomyopathies. For example, it is known, that DES mutations cause non-compaction cardiomyopathy (see 'Noncompaction cardiomyopathy is caused by a novel in-frame desmin (DES) deletion mutation within the 1A coiled-coil rod segment leading to a severe filament assembly defect’ (2019). It is also known, that splice site mutations in DES cause cardiomyopathies (‘The Desmin Mutation DES-c.735G>C Causes Severe Restrictive Cardiomyopathy by Inducing In-Frame Skipping of Exon-3’ (2021)). Both points I would mention and discuss in the context of a comparison of LMNA and DES.
This section has been added to the discussion
However, I think that the authors can fix these points. Therefore, I suggest a minor revision. Good luck!
We would like to thank the reviewer for their careful review of our manuscript and for their valuable comments and suggestions. We appreciate their positive remarks and the opportunity to improve the pedigree, table, and text in order to make them more understandable for readers.
We are grateful for their attention to detail and their constructive feedback.
Reviewer 3 Report
Comments and Suggestions for Authors
Authors have performed an interesting study of a family diagnosed with overlapped phenotypes associated with pathogenic alterations in the LMNA gene. Some points should be clarified:
1.- please, complete text in the line 128.
2.- the LMNA gene should be written in italic.
3.- please add complete classification (if possible, add please also data concerning population frequency, novel, etc...) of the intronic variant according to ACMG/AMP guidelines.
4.- please indicate the genes analyzed and rare variants identified in any of these genes.
5.- please indicate if any consanguinity in the family.
6.- please modify the word "mutation" to "rare variant" or "pathogenic variant" due to "mutation" means any alteration in the whole genome.
Author Response
Authors have performed an interesting study of a family diagnosed with overlapped phenotypes associated with pathogenic alterations in the LMNA gene. Some points should be clarified:
Thank you for your interest in our research. We thank the Reviewer for the important comments that made our manuscript better.
1.- please, complete text in the line 128.
We apologize that the names of the subsections were not numbered and italicized. Fixed it. Highlighted in blue.
2.- the LMNA gene should be written in italic.
Fixed it.
3.- please add complete classification (if possible, add please also data concerning population frequency, novel, etc...) of the intronic variant according to ACMG/AMP guidelines.
This information was added by us to the discussion section (highlighted in blue).
4.- please indicate the genes analyzed and rare variants identified in any of these genes.
Due to the characteristic clinical manifestations (age, nature of the lesion, pattern of affected muscles, and symptoms of lipodystrophy), as well as the availability of information about the autosomal dominant inheritance pattern of the condition within the family, a targeted analysis of the LMNA gene was initiated as part of the diagnostic process. No other rare LMNA variants were identified during this analysis.
5.- please indicate if any consanguinity in the family.
There were no consanguineous marriages in the family. Information added to the manuscript (highlighted in blue)
6.- please modify the word "mutation" to "rare variant" or "pathogenic variant" due to "mutation" means any alteration in the whole genome.
Fixed it.
Round 2
Reviewer 1 Report
Comments and Suggestions for Authors
Thank you for the detailed point-to-point response to my comments and for addressing these in the revised version of the manuscript. I have no further comments.
Reviewer 3 Report
Comments and Suggestions for Authors
No comments